# Impact of agile management on project performance: Evidence from I.T sector of Pakistan

Umer Muhammad[1]*, Tahira Nazir[1], Najam Muhammad[2], Ahsen Maqsoom[3], Samina Nawab[1], Syeda Tamkeen Fatima[1], Khuram Shafi[1], Faisal Shafique Butt[4]

1 Department of Management Science, COMSATS University Islamabad, Wah Cantt, Pakistan, 2 Department of Management Science, Riphah International University, Rawalpindi, Pakistan, 3 Department of Civil Engineering, COMSATS University Islamabad, Wah Cantt, Pakistan, 4 Department of Computer Science, COMSATS University Islamabad, Wah Cantt, Pakistan

* umer_stu@ciitwah.edu.pk

**Data Availability Statement:** All relevant data are within the paper and its Supporting information files.

## Abstract

Over the past several years, global project management teams have been facing dynamic challenges that continue to grow exponentially with the increasing number of complexities associated with the undertaken tasks. The ever-evolving organizational challenges demand project managers to adapt novel management practices to accomplish organizational goals rather than following traditional management practices. Considering which, the current study aims to explain the effect of agile management practices upon project performance directly as well as while being mediated through project complexity. Furthermore, the afore-mentioned mediatory relationship is evaluated in terms of the moderating effect of leadership competencies. The current study utilized the survey approach to collect the data from registered I.T firms deployed in the potential metropolitans of each province of Pakistan including, Peshawar, Islamabad, Lahore, Sialkot, Faisalabad, Hyderabad, Sukkur, and Karachi. A total of 176 responses were utilized for statistical evaluations. As result, it was observed that the negative influence anticipated by project complexity on project performance was compensated by the agile management practices. Further, the leadership competencies played a pivotal role in managing project complexity while implementing agile management practices and therefore enhancing project performance. The current study abridges the potential knowledge gap conceptually by evaluating the direct impact of agile management upon project performance while considering all of its aspects, exploring the mediatory role of project performance and evaluating the moderating role of leadership competencies in attaining optimum project performance. In contextual terms, the current study fills the knowledge gap by gauging the implications of agile management practices within the I.T sector of Pakistan. The results of the current study can be a potential guide for both the academicians and the industry professionals.

**Funding:** The authors received no specific funding for this work.

**Competing interests:** The authors have declared that no competing interests exist.

## Introduction

The agile management approach in terms of project development process remains rather a novel practice for most of the organizations of today to adapt and practice. Regardless, recent studies have indicated that organizations around the globe considering their long terms benefits are adapting the agile management practices more, in comparison to the traditionally followed waterfall management practices; especially in the IT sector. Research so far has highlighted the relevance of the agile management practices as well as has justified its constructive impact on the performance of an organization [1, 2]. In specific to the management trends being followed, a recent global report of PMI comprising opinion of 727 executive members deployed on 3,234 projects across Europe, Asia Pacific, North America, Latin American, Middle East, Africa, and Caribbean Regions, proposed the implementation of agile management practices as a potential reason to trigger organizational productivity. Therefore, signifying the impact of agile management practices upon the performance of the firms [3]. Moreover, another recent study conducted by Ambysoft indicated agile management practices to deliver a success rate of 55% in comparison to the waterfall management practices with a success rate of 29% only. The report further indicated that 36% of the projects completed under the agile management practices remained challenged and required limited fulfillment of constraints to accomplish the projects. In contrast, the waterfall management practices were credited 67% of the challenged projects. The study also revealed the agile management practices to be attributed with only a mere 3% of project failure rate [4]. Thus, justifying the constructive impact of agile management practices in terms of enhanced performance measures. Regardless, the precise study indicating the impact of implementing agile management practices upon the project performance while considering all of its related aspects is yet to be explored [5, 6]. Considering the potential research gap, the current study took into account of all relevant aspects of project performance including 'time', 'finances', 'magnitude of efforts', 'work environment moral', 'fulfillment of quality criterions' as well as the 'satisfaction of regarding stakeholders' and further observed the variation, in terms of the implementation of the agile management practices.

Considering the organizational accomplishment related aspect of the current research, the performance associated with the projects is often challenged by the magnitude of the complexity faced by the firms. Complexity, if not addressed timely can rile up to potential risks and consequently result in declined performance to a limit where it can jeopardize the existence of an organization itself. Considering which, research so far has indicted that implementation of relevant management practices can enable the mitigation of complexity associated to a project [7, 8]. As Sohi, Hertogh [9] in their recent study were able to justify the association of agile management practices with the abridged level of project complexity to some extent. It was further speculated by the researchers to enhance the project performance of any given firm. Therefore, to address the existing knowledge gap the current study took into account the mediating role of project complexity, to be able to analyze the direct impact of agility upon project complexity as well as the project performance. Moreover, justify the theorized impact of agility in terms of reduced project complexity and enhanced project performance.

Taking into account the managerial aspect of the current study, prior studies have indicated that the efficient and effective implementation of management practices for the most part has remained predominated by the human factor, and of which leadership competencies is of most vital consideration [10]. In various contexts, the effective implementation of leadership competencies has been found to have a significant impact on the overall organizational performance of any given firm [11, 12]. In relevance, a consolidated view of the implementation of leadership competencies to mitigate the organizational complexities and enhance performance measures is yet to be evaluated [13]. It is very much expectant of the agile management

practices to depict enhanced performance as a result of effective leadership competency mitigating the magnitude of dynamic organizational challenges. Considering which, the current study evaluated the moderating role of leadership competencies to observe the controlled impact of professional complexities and the delivered project performance. Therefore, filling in the existing conceptual knowledge gap indicated by prior researchers.

Furthermore, in specific to filling in the contextual research gap, the current study explored the implication of the targeted variables within the I.T sector of Pakistan, which itself has seen significant progression over the years.

The present study aims to accomplish the following research objectives:

RO1: *Determine the effect of agile management practices on project performance.*

RO2: *Evaluate the mediating role of project complexity between agile management practices and project performance.*

RO3: *Gauge the moderating role of leadership competencies between agile management practices and project complexity.*

The following sections of the study comprises of the detailed literature review of all the opted variables of the current study as well as their hypothetical development. Further, the methodological approach to collect the data from the targeted population is presented, which is then further statistically evaluated and explained in the results and analysis section. Followed to which, the deduction based upon the evaluated results are presented in the discussion. Lastly, the outcomes of the current research are deduced in the conclusion section.

## Literature review

**Agile management.** The concept of agile management got tossed in 1991 when the term agility was defined in a report by the Lacocca Institute, as "the ability to thrive in rapidly changing, fragmented markets". As the concept evolved, agility was redefined as, "the state or quality of being able to move quickly in an easy fashion". Therefore, for any firm labeled as agile is expectant to resolve unforeseeable challenges. Therefore, assuring the organizational sustainability in uncertain environments [14, 15]. The concept of agile management is multifaceted in nature and the remnants of its implementation have been observed across various disciplines over last few decades. Most early implementation of agile management practices was embraced by the manufacturing sector. At time, agility was defined as, "the capability of an organization to meet changing market requirements, maximize customer service levels and resultantly minimize the cost of goods" [16]. The agile management practices for a decade and more remained implemented within the manufacturing industry only [17]. It wasn't until the commercialization of the internet in 1995 when the agile management practices attained maturity in other industrial sectors as well, especially the software development [18]. To formalize the agility practices in terms of the software development process the OOPSA conference held in the same year played a momentous role when Ken Schwaber and Jeff Sutherland defined the cardinal principles for the implementation of agility on an organizational scale. Later, the agility saw minuscule implementation in the years to come, till 2001. It happened when various professionals, practitioners, and theorists came up with "Agile Manifesto", which was mutually signed and published on the internet. The manifesto challenged the implications of traditionally followed management practices onto the project-related outcomes with a higher level of uncertainties. Further, in addition to declaring the traditional management practices misaligned towards the dynamically natured projects, the report emphasized the induction of agile management practices in such environments. Thus, effectively managing organizational

objectives, minimizing project complexity, and delivering efficiency in terms of organizational performance [16, 19].

To understand what made the implementation of agile management practices a success in the software industry as well as its spread across the globe on the exponential rate in contrast to any other industry, one has to take into consideration the following factors on which the dynamics of agile management rely onto and further draw a comparison of them with the traditionally followed management practices [2, 20] (See Table 1).

The software industry has for most part evolved over the past 30 years. But the last decade has depicted a significant surge in the industry's growth and its respective performance. The reason justifying the phenomena has been the broader application of agile management practices, that replaced the traditionally followed management practices over time. The earlier research has justified the execution of agility in terms of ensuring enhanced performance, and also have supported the fact that implementation of agility is most suitable for the business environs that are dynamic in nature. Since, it has very vividly been observed that the implementation of software project development requires the dynamic implementation of operational measures as the problems are evolving real-time, which justifies the complexity associated with the software industry. Considering which, the software development sector is a perfect fit to adapt agile management practices [5].

Apart from the software products and services, one of the major parts of the project development process is the interaction between the stakeholders which plays a pivotal in determining the performance of the project. Considering which, Uludag, Kleehaus [22] and Hobbs and Petit [23] in their respective studies indicated that agile management practices allow organizations for its internal stakeholders to communicate freely as well as maintain a consistent stream of feedback from the external stakeholders. Thus, assuring the regarding organization to achieve optimal performance levels.

Considering the ability of agile management practices to enable its utilizers to accomplish projects in a dynamic environment and be able to deliver optimized performance while considering its respective dimensions i.e., competency, flexibility, quickness, and responsiveness, the current study took into account the implementation of agile management practices in relation to all the aspects of performance.

**H1: Agile management practices will significantly impact the project performance, in a positive manner.**

**Project complexity.**  Any given organization that functions onto various organizational factors either human or non-human operating in parallel to one another, is bound to face unexpected challenges to manage through and accomplish its goals. Considering which, the software industry has been the most critical one on the list [24]. It has been so because regardless of the business type, every operational entity is reliant on the software utilization either it is in form of communication, logistics, traveling, academia, and even fields as critical as healthcare. Therefore, justifying the software industry to be the one facing crucial levels of complexity [25].

Typically, for a large-scale operation with a higher magnitude of complexity, like software development, is often considered as a project rather than a routine-based operation/task, by most of the organizations. This demands a persistent application of relevant management practices under effective supervision to tackle the complexity.

For the successful accomplishment of a project, opting relevant management approach plays a pivotal role in tackling the complexities associated with the environment. Since only the right management approach can enable the managers to make correct calculations to

**Table 1. Management practices & suitable conditions [20, 21].**

| Conditions | Agile Management | Waterfall Management |
|---|---|---|
| **Business Environ** | The client's preferences and respective solutions are dynamic. | The business environ is stable and foreseeable. |
| **Project's Ownership** | The ownership is shared equally between all the members of the team as well as its associated stakeholders. | The project manager is considered as the owner of the project in terms of operations, until its accomplishment. |
| **Working Style** | Free flow of communication is encouraged so that everyone can float their ideas. | Hierarchical management is implemented. Which allows the implementation of directs from the top management only. |
| **Client's Involvement** | The client's requirements evolve as the project proceeds. | The client's demands remain stable throughout the project. |
| **Project's Preplanning** | The preplanning phase remains active throughout every step of the project. | The preplanning is conducted once before the initiation of every major phase of the project. |
| **Project's Planning** | The plan is delivered to the client in distributed chunks. | A one-time plan is delivered to the client, along with a negligible margin of change. |
| **Innovation Type** | The solutions associated with the designated problem are not clear. Because of which, the product specification may alter over time. | The level of innovation is not groundbreaking. Solutions are very clear. Detailed plans along with the project's specifications are available to proceed with. |
| **Modular Approach** | The work can be broken down into independently working modules, which can be improved by iterative updating. | The testing of the product cannot begin until the project is accomplished in terms of the developmental phase. |
| **Project's Deliverance** | The project is delivered as well as parallelly evolved in terms of incremental deliverance. | The project is delivered on a one-time basis. |
| **Change Affordability** | Late alterations to the project can be accommodated and will not incur significant losses. | Late alterations to the project can either be very expansive or impossible to accommodate. |
| **Managing Changes** | Changes can be made in any module at any point in the development phase, with a greater amount of flexibility. | A top-down pattern in managing changes is observed since the development of the project is carried on from one phase leading to another. |
| **Interim Errors** | Errors are known to provide insight to the possible improvements to the project and are considered as an asset in terms of knowledge. | Errors may result in catastrophic situations for the organization, working on the project. |

allocate the right percentage of resources to the right places at the right time. Moreover, the application of a relevant management approach enables the mitigation of risk and the magnitude of projected losses [2, 26].

Prior studies have indicated a directly proportionate relationship between the complexity and the respective performance of an organization and the projects associated with it. This suggests that if the complexities associated with any given project are not handled effectively on time, are probable to cause an escalation in the level of hindrances associated with the project and may even result in failure of the project itself [27, 28].

Project complexity attributed to any given project is determined upon the variation in the number of tasks, their respective types, individuals deployed, and numerous other considerations. Considering which, effective prioritization of the entities involved, and the correct allocation of resources is necessary. All of which is only possible through the application of the relevant management approach [8].

Past decades have seen an evolution in terms of management practices and their respective application. Which have encouraged both academia as well as practitioners to extend the knowledge upon. As a matter of fact, among the two widely practiced project development management approaches i.e. waterfall and agile, it is the agile management approach that has proved itself to be more efficient to accomplish projects, across the world [29].

Considering which, Zhu and Mostafavi [8] in their study indicated the ability of agile management practices to manage through complex settings more effectively and efficiently. Thus, suggesting to lead the project towards better performance. Moreover, in another study Maylor and Turner [27] highlighted the aspect of stakeholder's involvement in the development process, which justified the mitigation of project complexity to a greater extent. As agile

management encourages the internal stakeholders of the project to seek continuous feedback from one another as well as from the clients throughout the process. Doing so reduces the amount of ambiguity from the development phase as much as possible and induce desired changes along the process. Thus, the finished project is much more of a reflection of the client's expectations and assurance of enhanced performance. Moreover, in specific to the software industry the nature of projects is bound to change much more rapidly than any other industry, which classifies the software industry with the highest level of complexity attributed to it. For its resolution, the agile management approach suggests breaking down of complex scenarios into smaller tasks with reduced complexity. Thus, resulting in the effective and focused application of management practices, which would further result in mitigation of complexity associated with the project as well as elevated project performance [18, 27].

Considering, the ability of agile management practices to mitigate the magnitude of complexity associated with the project and enhance the chances of the performance associated to the regarding project accomplish projects in a dynamic environment, the current study took into account the direct implementation of agile management practices in relation to the diminished project complexity. Moreover, the project complexity was evaluated in terms of a mediator.

**H2: Agile management practices will significantly impact the project complexity, in a negative manner.**

**H3: Project complexity will significantly impact the project performance, in a negative manner.**

**H4: Project complexity will significantly mediate the relationship between agile management practices and project performance.**

**Leadership competencies.** The opting of management practices is not enough for an organization to function properly. Rather it is the effective implementation of those defined policies that ensure the magnitude of performance delivered and subsequently the overall sustainability of an organization. For which, it is the human factor in terms of leadership, within an organization that contributes the most towards it. This is where leadership and its respective competencies come into play. Andriukaitienė, Voronkova [30] in their study defined project manager competence as a combination of knowledge (qualification), skills (ability to do a task), and core personality characteristics (motives, traits, self-concepts) that lead to superior results.

In the project management literature, few topics are too frequently discussed yet are very rarely agreed upon; such as the aspect of project performance [2]. The last two decades have extended the scope of project performance far beyond the measures of cost, time, and functionality. The project performance measures of today demand to fulfill the satisfaction criterion of the stakeholder associated with the given project, attainment of business/ organizational goals, product success, and development of the team involved. All of which is very much reliant upon the effectiveness of the implied organizational practice under human supervision [31]. Refereed to which, Maqbool, Sudong [32] in their study identified the possible shortcoming that may hinder the performance associated to any given project. The findings identified the hindering effects as the ineffective management practice observed in the planning, organization, and controlling of the project. Furthermore, Alvarenga, Branco [33] identified various performance measures associated with well-executed projects. Overall, the findings reflected the leadership competency in terms of maintaining effective communication and problem solving resulted in enhanced project performance. While, the absence of

leadership competency in terms of inadequate administration/supervision, human skills, and emotional influencing skills (IQ & EQ) resulted in declined performance or even failure in some cases. Ahmed and Anantatmula [34] in his study suggested that the manager's perception of performance and belief in his/her ability can play a significant role in determining the performance delivered. Thus, deeming the leadership competency to play a pivotal role in the accomplishment of a project. Akin to which, Turner came up with the seven forces model to define the factors influencing the project's performance. The model highlights the people as the cardinal force to drive the project towards accomplishment; which is only possible through leadership competencies, teamwork, and industrial relations. Hassan, Bashir [35] in their studies brought up the subject that despite the vast research on the project performance and its related measures the organizations still fail to satisfy its stakeholders. It was because most of the research done so far was considering time, cost, and quality as the only measure to determine the project performance delivered. Hassan, Bashir [35] and Maqbool, Sudong [32] indicated the criticality of including the human factor in terms of leadership competence/ability to determine the performance of the project. Zuo, Zhao [36] and Gunter [37] as well in their studies reviewed the impact of leadership's competence and style to determine the project's outcomes and concluded the fact that the existing literature has for most part overlooked the impact of leadership competence on the project's performance. Therefore, to evaluate the controlling effect of leadership competency to observe change in the magnitude of the performance delivered, the current study proposed the following hypothesis (See Fig 1).

**H5: Leadership competencies will significantly impact the project performance, in a positive manner.**

**H6: Leadership competencies will significantly moderate the relationship between project complexity and project performance.**

## Research methodology

The current study implemented a cross-sectional quantitative approach to make a deduction regarding the variables under consideration. The population opted for the current study was comprised of the registered IT firms deployed across Pakistan, with a total count of 1800 [38]. The sampling was further conducted through a cluster approach to determine the targeted respondents. The country was considered as being divided into five clusters, which are also its provinces i.e. Khyber Pakhtunkhwa, Gilgit Baltistan, Punjab, Sindh, and Baluchistan. The sample count of each cluster was proportionately determined while considering the number of IT firms deployed in each province. This approach enabled determining the unbiased opinion of the general population, rather than the opinion of respondents associated with a specific cluster; overshadowing the other clusters [39]. Conclusively a sample size of 176 responding firms was evaluated through the sample size formulation commended by [40, 41] (See Eq 1).

$$\text{Sample Size} = \frac{\frac{z^2 * p(1-p)}{e^2}}{1 + \left(\frac{z^2 * p(1-p)}{e^2 N}\right)} \tag{1}$$

Where (z) is the representative of the corresponding z-score, (e) is the margin of error, and (N) is the population size. Whereas the confidence interval taken for the study was 99%.

The survey questionnaire was composed of 48 items in total. To determine the application of agile management practices on the organizational level a 20 relevant items were adapted from the scale developed by Zhang and Sharifi [42]. The scale itself was based upon four dimensions i.e. ability, flexibility, quickness, and responsiveness. To determine the leadership

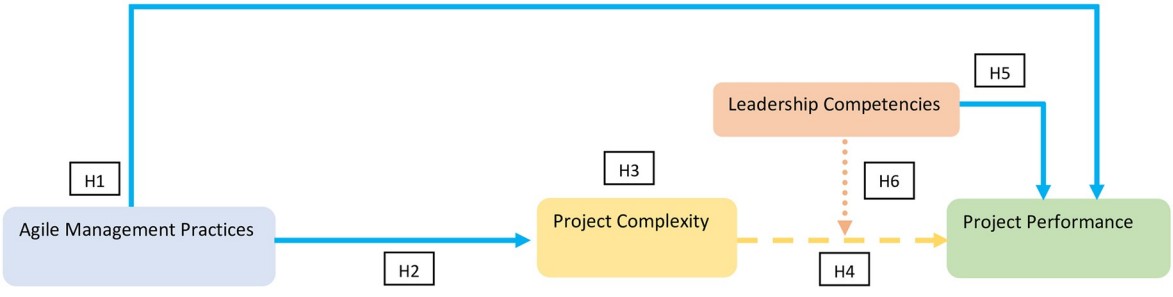

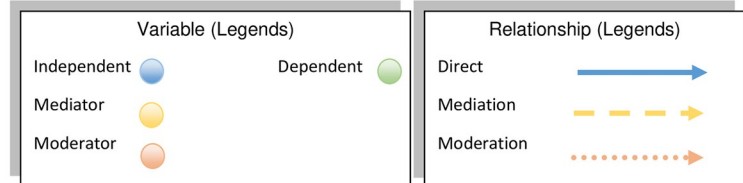

**Fig 1. Conceptual model.**

competencies of managers on various hierarchical levels of an organization, an 10 items were adapted from the scale developed by Chung-Herrera, Enz [43]. The scale was composed of 8 unique dimensions i.e. self-management, strategic positioning, implementation, critical thinking, communication, interpersonal, leadership, and industry knowledge. To determine the overall magnitude of complexity associated with the project under study, 12 items were adapted from the scale developed by Xia and Lee [44]. To determine the overall performance of the undertaken projects, a 6 items scale developed by Yusuf, Sarhadi [45] was utilized in the current study. The responses were recorded upon the 5-Point Likert scale, which had (1) to refer to "Strongly Disagree" up to (5) referring to "Strongly Agree" [46].

The current study included the opinion of the respondents recorded in terms of quantitative scale. During the data collection process, no confidential information (personal/organizational) was inquired about. Also, the presented research did not categorize the involved workers in terms of race/ethnicity, age, disease/disabilities, religion, sex/gender, sexual orientation, or other socially constructed groupings. Therefore, COMSATS University Islamabad's Ethics Review Committee declared the current study exempted from the requirement of consent from the respondents. Considering which, a total of 250 questionnaires were disseminated to survey the professionals of the Pakistani IT industry. By the end of the survey process, a total of 190 responses got collected. Thus, the overall response rate of the study was 76%. Further, 7% of the responses were discarded as a result of being incomplete or erroneous. Since both incomplete or redundant data can affect the results adversely [47]. Followed to the collection of data the next phase demanded the application of appropriate statistical tools and respective data analysis techniques to make deductions regarding the objectives of the study. For which the current study utilized the SmartPLS GmbH's SMART Partial Least Squares

(SMART PLS 3.0) to analyze the dataset. Various studies in recent years have utilized a similar tool and respective techniques to analyze the data and make respective deductions [48, 49].

## Statistical results & analysis

To begin with, the information was gauged to assess the instrument's reliability and validity. Further, the instrument's fitness was evaluated in terms of factor loadings. The results identified few unfit components associated with the variables under study. Suggested to which, the identified unfit components of the hypothesized model were then removed. Followed by which, the information was evaluated to gauge the direct and indirect effects of variables, in alignment with the hypothesized model. Finally, the hypothesized model was concluded upon the evaluation of the total impact of the predictor variables upon the dependent variable [50, 51].

### Demographical classification

The respondents of the study had variating attributions associated with them in terms of demographics. The current study classified the respondents in terms of age, tenure of employment, sector of employment, the status of employment, and the geographical location of their organization.

As a response to which 63.6 percent of employees were aged between 20–29 years, 21.6 percent were aged between 30–39 years, 10.8 percent were aged between 40–49 years and 4.0 percent were aged 50 years or above.

In specific to the tenure of employment or the managerial experience, 27.8 percent of respondents had an experience of less than 1 year, 20.5 percent had experience ranged between 1–2 years, 19.3 percent had experience ranged between 2–5 years, 9.1 percent had experience ranged between 5–10 years and, 23.3 percent had an experience of 10 years or over.

In terms of the employment sector, 53.9 percent of the individuals were employed in the public sector. While 46.1 percent of the individuals were employed in the private sector.

In terms of the geographical placement of the surveyed organizations, 12.5 percent of the firm were deployed in the Khyber Pakhtunkhwa and Gilgit Baltistan, 50 percent of the firm were deployed in Punjab, 25 percent of the firm were deployed in the Sindh and, 12.5 percent of the firm were deployed in the Balochistan. Thus, deeming the study to utilize the equivalently proportionate responses from each province, that were aligned with the proportion of firms in each province, nationwide.

### Structural equation modeling

Structural equation modeling is a multivariate based statistical evaluation approach that is utilized to determine structural associations between the components of a hypothesized model [52, 53]. The adapted approach is a combination of factor analysis and multiple- regression analysis. The current study took a two-stage approach to conduct SEM. The first stage involved the application of confirmatory factor analysis (CFA), which justified the consistency of the research instrument and its associated components/items. Followed by which, the research instrument was tested for its respective reliability and validity in the first stage of SEM, as commended by prior research [53]. The second stage of SEM involved the evaluation of measuring the magnitude of impact existent between the observed and latent variables under discussion. Which were further justifies in terms of their significance and their respective relevance in alignment to the hypothesized relationships [54].

**SEM (stage 1).** To begin with, the first stage of the SEM tested the measurement model for its reliability, validity (convergent, discriminant), and consistency to the components

towards the research instrument, utilizing the CFA approach. CFA is a commended approach to test adapted research instruments for their consistency [49, 55].

**Instrument's reliability.** The reliability of a research instrument is its ability to give consistent results with negligible variation regardless of the environment it is utilized in. SEM utilizes Cronbach's Alpha as the criterion of reliability associated with a research instrument. For a research instrument and its respective components to be reliable the value of Cronbach's Alpha is commended to be higher than 0.70 [56]. Keeping that in view, the values of Cronbach's Alpha associated with all the variables under study were above 0.70 (See Table 2). Thus, deeming the respective research instrument to be reliable.

**Instrument's validity (convergent).** The validity of a research instrument is defined as its ability to measure the phenomena that it is supposed to measure. There are two types of validity i.e. convergent and discriminant [57, 58]. The convergent validity associated with a research instrument is the measure to determine the relatability of research items to their respective variable. SEM utilizes Average Variance Extracted (AVE) as the criterion of validity associated with a research instrument. For a research instrument and its respective components to be convergently valid, the value of AVE is commended to be higher than 0.5 [49, 59]. Keeping, that in view the values of AVE associated with all the variables under study were above 0.5 (See Table 3). Thus, deeming the respective research instrument to be convergently valid.

**Instrument's validity (discriminant).** The discriminant validity associated with a research instrument is the measure to determine the magnitude of dissimilarity of research items associated with a variable towards the research items of the rest of the variables under study. SEM utilizes Fornell-Larcker Criterion as the criterion of discriminant validity associated with a research instrument. For a research instrument and its respective components to be discriminately valid, the correlative value of Fornell-Larcker Criterion of a variable with its components is commended to be higher than the correlative value of other variables in the study [48, 49]. Keeping, that in view the values of the Fornell-Larcker Criterion associated with all the variables under study were comparatively higher than the correlative values of other variables in the study (See Table 4). Thus, deeming the respective research instrument to be discriminately valid.

Another measure to determine, the discriminant validity associated to a research instrument is the Cross Loadings. For a research instrument and its respective components to be discriminately valid, the correlative values of Cross Loadings of the items of a variable are commended to be higher than the correlative values of similar items with other variables in the study [49]. Keeping, that in view the values of Cross Loadings associated to all the items of the variables under study were comparatively higher than the correlative values of similar items with rest of the variables in the study (See Table 5). Thus, deeming the respective research instrument to be discriminately valid.

Lastly, in terms of evaluating the discriminant validity, the Heterotrait-Monotrait Ratio (HTMT) is considered as the most precise measurement. HTMT is based upon a higher level of specificity that is ranged between the measurement precision of 97%-99%. On the contrary, the measures of Cross Loadings followed by the Fornell-Larcker Criterion can only depict a measurement precision ranged between 0.00%-20.82% [49, 60]. In terms of HTMT, for a research instrument to be valid, the correlational terms must be valued lower than the 0.90. Keeping that in view, the correlation values associated with all the variables were below 0.90 (See Table 6). Thus, deeming the respective research instrument to be discriminately valid.

**Multi-collinearity.** Multi-Collinearity is the state of higher correlation existent between the variables and the indicators associated with them. Which can further lead to unreliable statistical projections and inferences. To test a variable and its respective indicators for collinearity, the proposed criterion of VIF is followed. The referred criterion suggests for all the

**Table 2. Cronbach's alpha.**

|  | Cronbach's Alpha | rho_A | Composite Reliability |
|---|---|---|---|
| **Agile Management** | 0.933 | 0.934 | 0.940 |
| **Leadership Competencies** | 0.904 | 0.909 | 0.921 |
| **Project Complexity** | 0.867 | 0.867 | 0.889 |
| **Project Performance** | 0.891 | 0.894 | 0.917 |

**Table 3. Composite validity (AVE).**

|  | Average Variance Extracted (AVE) |
|---|---|
| **Agile Management** | 0.542 |
| **Leadership Competencies** | 0.538 |
| **Project Complexity** | 0.500 |
| **Project Performance** | 0.648 |

**Table 4. Fornell-Larcker criterion.**

|  | Agile Management | Leadership Competencies | Project Complexity | Project Performance |
|---|---|---|---|---|
| **Agile Management** | 0.765 |  |  |  |
| **Leadership Competencies** | 0.693 | 0.763 |  |  |
| **Project Complexity** | 0.703 | 0.601 | 0.634 |  |
| **Project Performance** | 0.742 | 0.753 | 0.548 | 0.805 |

indicators of the regarding variable to have a VIF value lower than 5 to be fit in terms of collinearity measure [48]. Keeping that in view all the indicators associated with the variables under study were found to have VIF value under 5 (See Table 7).

**Factor loadings.** Followed to fulfilling the criterion of the research instrument's reliability and validity the respective components must fulfill the criterion of factor analysis that is measured in terms of Factor Loadings. Factor Loadings are determinant of the variability and correlation associated with the items of the observed variables under study. For an item associated with a variable to fulfill the Factor Loading criterion, must be valued above 0.7 [61, 62]. In comparison to which, selective items associated with agile management (AM13) and project complexity (PC2, PC4) were found below the commended threshold value (See Table 8). Thus, these items were removed from the measurement model, to enhance the overall fit.

**SEM (stage 2).** After the deletion of unfit components of the measurement model, the second stage involved the reassessment of the measurement model. The model was retested in terms of Factor Loadings, which depicted all of the values to be ranged above the minimum threshold of 0.70 [62] (See Table 9).

**Path coefficients.** After conforming to the component fitness criterion, the structural model was evaluated in terms of the magnitude of the effect the observed variables had on the latent variables. The said magnitude was evaluated utilizing the measure of Path Coefficients. The value associated to the measure of path coefficient varies between ±1, which suggests the positive and negative relationship between the variables under consideration [48, 63, 64]. The effect of agile management practices over the project performance was valued at 0.473. The effect of agile management practices over the project complexity was valued as 0.703. The

**Table 5. Cross loadings.**

|        | Agile Management | Leadership Competencies | Project Complexity | Project Performance |
|--------|------------------|-------------------------|--------------------|---------------------|
| AM1    | 0.782            | 0.563                   | 0.564              | 0.557               |
| AM2    | 0.714            | 0.487                   | 0.470              | 0.488               |
| AM3    | 0.750            | 0.461                   | 0.454              | 0.564               |
| AM4    | 0.696            | 0.385                   | 0.479              | 0.485               |
| AM5    | 0.677            | 0.434                   | 0.411              | 0.546               |
| AM6    | 0.675            | 0.467                   | 0.424              | 0.505               |
| AM7    | 0.662            | 0.604                   | 0.471              | 0.594               |
| AM8    | 0.763            | 0.490                   | 0.545              | 0.519               |
| AM9    | 0.670            | 0.433                   | 0.463              | 0.485               |
| AM10   | 0.616            | 0.368                   | 0.353              | 0.439               |
| AM11   | 0.621            | 0.415                   | 0.503              | 0.429               |
| AM12   | 0.630            | 0.419                   | 0.481              | 0.476               |
| AM13   | 0.468            | 0.304                   | 0.414              | 0.358               |
| AM14   | 0.678            | 0.495                   | 0.430              | 0.547               |
| AM15   | 0.597            | 0.420                   | 0.520              | 0.372               |
| AM16   | 0.618            | 0.452                   | 0.451              | 0.494               |
| AM17   | 0.648            | 0.427                   | 0.476              | 0.428               |
| AM18   | 0.648            | 0.518                   | 0.454              | 0.454               |
| AM19   | 0.670            | 0.507                   | 0.468              | 0.552               |
| AM20   | 0.656            | 0.523                   | 0.413              | 0.567               |
| LC1    | 0.571            | 0.753                   | 0.432              | 0.569               |
| LC2    | 0.640            | 0.808                   | 0.500              | 0.638               |
| LC3    | 0.564            | 0.735                   | 0.452              | 0.646               |
| LC4    | 0.564            | 0.778                   | 0.491              | 0.570               |
| LC5    | 0.362            | 0.659                   | 0.344              | 0.390               |
| LC6    | 0.497            | 0.717                   | 0.409              | 0.541               |
| LC7    | 0.472            | 0.734                   | 0.470              | 0.539               |
| LC8    | 0.437            | 0.703                   | 0.414              | 0.539               |
| LC9    | 0.396            | 0.697                   | 0.367              | 0.483               |
| LC10   | 0.508            | 0.736                   | 0.496              | 0.543               |
| PC1    | 0.563            | 0.536                   | 0.628              | 0.456               |
| PC2    | 0.442            | 0.417                   | 0.575              | 0.394               |
| PC3    | 0.517            | 0.464                   | 0.659              | 0.489               |
| PC4    | 0.488            | 0.334                   | 0.590              | 0.341               |
| PC5    | 0.507            | 0.455                   | 0.625              | 0.395               |
| PC6    | 0.418            | 0.337                   | 0.636              | 0.304               |
| PC7    | 0.453            | 0.314                   | 0.697              | 0.289               |
| PC8    | 0.365            | 0.253                   | 0.650              | 0.238               |
| PC9    | 0.322            | 0.264                   | 0.671              | 0.201               |
| PC10   | 0.356            | 0.364                   | 0.648              | 0.350               |
| PC11   | 0.402            | 0.329                   | 0.632              | 0.281               |
| PC12   | 0.318            | 0.315                   | 0.587              | 0.212               |
| PP1    | 0.560            | 0.521                   | 0.383              | 0.762               |
| PP2    | 0.547            | 0.560                   | 0.442              | 0.792               |
| PP3    | 0.601            | 0.591                   | 0.511              | 0.806               |
| PP4    | 0.581            | 0.657                   | 0.398              | 0.807               |
| PP5    | 0.629            | 0.656                   | 0.448              | 0.819               |

*(Continued)*

**Table 5.** (Continued)

|  | Agile Management | Leadership Competencies | Project Complexity | Project Performance |
|---|---|---|---|---|
| **PP6** | 0.657 | 0.636 | 0.460 | 0.843 |

AM: Agile Management, LC: Leadership Competencies, PC: Project Complexity, PP: Project Performance.

effect of leadership competencies over the project performance was valued at 0.664. Lastly, the effect of project complexity over the project performance was valued at 0.149. The evaluated effects were further justified in terms of the level of significance attributed to them i.e. *p-value* ≤ 0.05. Since all the results fulfilled the significance criterion, for which the evaluated effects were considered as accepted (See Table 10). Thus, justifying the following hypothesized relationships between the variables under study:

**Coefficient of determination ($r^2$).** Coefficient of Determination ($r^2$) is representative of the amount of variance the exogenous variable/s can cause in the associated endogenous variable/s. The value of the Coefficient of Determination *(r2)* varies between 0–1. The higher the value of $r^2$ the higher the magnitude of impact implied by the exogenous variables [65]. Keeping that in view, the exogenous variables of the study i.e. (Agile Management, Project Complexity, and Leadership Competencies) impacted the endogenous variable i.e. (Project Performance) with an $r^2$ valued at 0.582. Thus, justifying 58.20% of the variance explained (See Table 11).

**Effect size ($f^2$).** Effect Size ($f^2$) is representative of the magnitude of effect an exogenous variable can have on an endogenous variable. The respective magnitude of the effect is classified into three tiers. For a given relationship the values of Effect Size ($f^2$) ranged between 0.02–0.14 are attributed as a small effect. Likewise, values ranged between 0.15–0.35 are attributed as a medium effect, and values ranged 0.36 and above are attributed as a large effect [48, 51]. Keeping that in view, both the agile management and project complexity had a medium impact. While leadership competencies and project complexity had a large effect on their respective dependent variables. (See Table 12).

**Mediation analysis.** A mediatory variable of the study is known to add an explanation or justify the effect of an exogenous variable over an endogenous variable. The current study took project complexity as a mediator to explain the effect of agile management over the project performance. SmartPLS explains the mediation in terms of Indirect Effects and its respective significance [66, 67]. Keeping, that in view the hypothesized mediation was approved (See Table 13). Thus, accepting the following hypothesis:

**Moderation analysis.** A moderating variable of the study is known to control the magnitude of the effect of an exogenous variable over an endogenous variable. This effect can be tilted either positively or negatively in presence of a moderator. The current study took leadership competencies as a moderator to control the effect of project complexity over the project performance. SmartPLS explains the moderation in terms of inducing a product indicator

**Table 6. Heterotrait-monotrait ratio (HTMT).**

|  | Agile Management | Leadership Competencies | Project Complexity | Project Performance |
|---|---|---|---|---|
| **Agile Management** |  |  |  |  |
| **Leadership Competencies** | 0.744 |  |  |  |
| **Project Complexity** | 0.744 | 0.644 |  |  |
| **Project Performance** | 0.815 | 0.826 | 0.587 |  |

**Table 7. VIF.**

| Indicator | VIF |
|---|---|
| AM1 | 2.897 |
| AM2 | 2.306 |
| AM3 | 2.692 |
| AM4 | 2.074 |
| AM5 | 2.128 |
| AM6 | 2.033 |
| AM7 | 1.892 |
| AM8 | 2.332 |
| AM9 | 2.062 |
| AM10 | 1.606 |
| AM11 | 1.632 |
| AM12 | 1.932 |
| AM13 | 1.743 |
| AM14 | 2.040 |
| AM15 | 1.666 |
| AM16 | 1.881 |
| AM17 | 1.970 |
| AM18 | 2.037 |
| AM19 | 2.050 |
| AM20 | 2.088 |
| LC1 | 2.567 |
| LC2 | 2.717 |
| LC3 | 2.153 |
| LC4 | 2.381 |
| LC5 | 1.737 |
| LC6 | 1.781 |
| LC7 | 1.866 |
| LC8 | 1.972 |
| LC9 | 2.107 |
| LC10 | 2.041 |
| PC1 | 1.602 |
| PC2 | 1.407 |
| PC3 | 1.755 |
| PC4 | 1.538 |
| PC5 | 1.763 |
| PC6 | 1.841 |
| PC7 | 2.000 |
| PC8 | 1.995 |
| PC9 | 2.299 |
| PC10 | 2.085 |
| PC11 | 2.204 |
| PC12 | 1.898 |
| PP1 | 1.920 |
| PP2 | 2.046 |
| PP3 | 2.051 |
| PP4 | 2.032 |
| PP5 | 2.254 |
| PP6 | 2.492 |

AM: Agile Management, LC: Leadership Competencies, PC: Project Complexity, PP: Project Performance.

**Table 8. Factor loadings.**

|  | Agile Management | Leadership Competencies | Project Complexity | Project Performance |
|---|---|---|---|---|
| AM1 | 0.782 | | | |
| AM2 | 0.714 | | | |
| AM3 | 0.750 | | | |
| AM4 | 0.796 | | | |
| AM5 | 0.777 | | | |
| AM6 | 0.775 | | | |
| AM7 | 0.762 | | | |
| AM8 | 0.763 | | | |
| AM9 | 0.770 | | | |
| AM10 | 0.716 | | | |
| AM11 | 0.721 | | | |
| AM12 | 0.730 | | | |
| AM13 | 0.668 | | | |
| AM14 | 0.778 | | | |
| AM15 | 0.797 | | | |
| AM16 | 0.718 | | | |
| AM17 | 0.748 | | | |
| AM18 | 0.748 | | | |
| AM19 | 0.770 | | | |
| AM20 | 0.756 | | | |
| LC1 | | 0.753 | | |
| LC2 | | 0.808 | | |
| LC3 | | 0.735 | | |
| LC4 | | 0.778 | | |
| LC5 | | 0.759 | | |
| LC6 | | 0.717 | | |
| LC7 | | 0.734 | | |
| LC8 | | 0.703 | | |
| LC9 | | 0.797 | | |
| LC10 | | 0.736 | | |
| PC1 | | | 0.728 | |
| PC2 | | | 0.575 | |
| PC3 | | | 0.759 | |
| PC4 | | | 0.590 | |
| PC5 | | | 0.725 | |
| PC6 | | | 0.736 | |
| PC7 | | | 0.797 | |
| PC8 | | | 0.750 | |
| PC9 | | | 0.771 | |
| PC10 | | | 0.648 | |
| PC11 | | | 0.632 | |
| PC12 | | | 0.787 | |
| PP1 | | | | 0.762 |
| PP2 | | | | 0.792 |
| PP3 | | | | 0.806 |
| PP4 | | | | 0.807 |
| PP5 | | | | 0.819 |
| PP6 | | | | 0.843 |

AM: Agile Management, LC: Leadership Competencies, PC: Project Complexity, PP: Project Performance.

**Table 9. Factor loadings.**

|  | Agile Management | Leadership Competencies | Project Complexity | Project Performance |
|---|---|---|---|---|
| **AM1** | 0.781 |  |  |  |
| **AM2** | 0.727 |  |  |  |
| **AM3** | 0.753 |  |  |  |
| **AM4** | 0.798 |  |  |  |
| **AM5** | 0.782 |  |  |  |
| **AM6** | 0.781 |  |  |  |
| **AM7** | 0.765 |  |  |  |
| **AM8** | 0.765 |  |  |  |
| **AM9** | 0.769 |  |  |  |
| **AM10** | 0.715 |  |  |  |
| **AM11** | 0.713 |  |  |  |
| **AM12** | 0.718 |  |  |  |
| **AM14** | 0.771 |  |  |  |
| **AM15** | 0.792 |  |  |  |
| **AM16** | 0.726 |  |  |  |
| **AM17** | 0.751 |  |  |  |
| **AM18** | 0.753 |  |  |  |
| **AM19** | 0.777 |  |  |  |
| **AM20** | 0.760 |  |  |  |
| **LC1** |  | 0.753 |  |  |
| **LC2** |  | 0.808 |  |  |
| **LC3** |  | 0.735 |  |  |
| **LC4** |  | 0.778 |  |  |
| **LC5** |  | 0.659 |  |  |
| **LC6** |  | 0.717 |  |  |
| **LC7** |  | 0.734 |  |  |
| **LC8** |  | 0.703 |  |  |
| **LC9** |  | 0.797 |  |  |
| **LC10** |  | 0.736 |  |  |
| **PC1** |  |  | 0.704 |  |
| **PC3** |  |  | 0.729 |  |
| **PC5** |  |  | 0.718 |  |
| **PC6** |  |  | 0.752 |  |
| **PC7** |  |  | 0.732 |  |
| **PC8** |  |  | 0.790 |  |
| **PC9** |  |  | 0.710 |  |
| **PC10** |  |  | 0.767 |  |
| **PC11** |  |  | 0.782 |  |
| **PC12** |  |  | 0.721 |  |
| **PP1** |  |  |  | 0.762 |
| **PP2** |  |  |  | 0.792 |
| **PP3** |  |  |  | 0.806 |
| **PP4** |  |  |  | 0.807 |
| **PP5** |  |  |  | 0.819 |
| **PP6** |  |  |  | 0.843 |

AM: Agile Management, LC: Leadership Competencies, PC: Project Complexity, PP: Project Performance.

**Table 10. Path coefficients.**

| | Original Sample (O) | Sample Mean (M) | T Statistics (\|O/STDEV\|) | P Values |
|---|---|---|---|---|
| **Agile Management > Project Performance** | 0.460 | 0.465 | 5.848 | 0.000 |
| **Agile Management > Project Complexity** | -0.703 | -0.714 | 19.007 | 0.000 |
| **Leadership Competencies > Project Performance** | 0.664 | 0.669 | 13.639 | 0.000 |
| **Project Complexity > Project Performance** | -0.149 | -0.149 | 2.283 | 0.023 |

term in the structural model and its respective significance [68]. Keeping, that in view the hypothesized moderation was approved (See Table 14). Thus, accepting the following hypothesis:

**Results summary.** The proposed hypotheses for the current study were accepted while considering their significance. The respective summary is depicted in the following Table 15.

## Discussion

To begin with, the first research hypothesis stated, "Are the agile management practices a significant predictor of project performance?". Keeping that in view, the current study depicted a significantly positive influence of implementing agile management practices onto the overall performance of the projects undertaken. This suggests, that resolving a project into smaller functional proportions and responding timely is a commendable approach to enhance the performance of the undertaken projects.

Furthermore, the statistical findings in accordance with the dimensions of the agile management the significance of the relationship emphasized that an organization must undertake only the projects that it is competent enough to accomplish. Moreover, for a project that is undertaken, must be resolved down to work units that can be matched with the competency level of the employed individual. This would enable them to achieve the targeted goals with fewer hurdles faced along the process. Similar results were concluded by Alvarenga, Branco [33] in their study conducted on 257 project managers; each having an extensive experience of over 10 years. As it was indicated that it is the competency associated to the employed individuals in an organization that assures the efficient and effective execution of organizational task and result in accomplishment of the undertaken projects. Followed to which, agile management commends the adaption of flexibility in the project development process that allows the project team to incorporate the changes more easily than the traditional implementation of the projects. Similarly, the loss incurred during the development process is relatively less. Since the failure is often observed in one or a few modules at a time, which doesn't affect the rest of the development process in any way. Most importantly, agile management is most responsible for responding quickly to the areas of projects that demand prioritized completion or technical handling. The respective findings were found in alignment to the study conducted by Serrador and Pinto [5] on 1002 projects deployed across various nations, that depicted a similar notion of a positive impact of implementing agile management to attain enhanced organizational outcomes. In another mixed-mode study conducted by Drury-Grogan [69] on various teams

**Table 11. Coefficient of determination ($r^2$).**

| | R Square | R Square Adjusted |
|---|---|---|
| **Project Complexity** | 0.494 | 0.491 |
| **Project Performance** | 0.582 | 0.577 |

**Table 12. Effect size ($f^2$).**

|  | Agile Management | Leadership Competencies | Project Complexity | Project Performance |
|---|---|---|---|---|
| **Agile Management** |  |  | 0.975 | 0.250 |
| **Leadership Competencies** |  |  |  | 0.673 |
| **Project Complexity** |  |  |  | 0.034 |
| **Project Performance** |  |  |  |  |

utilizing agile tools in the I.T sector as well suggested that application of the referred tools resulted in enhancing the success associated with the regarding projects.

The second research hypothesis stated, "Are the agile management practices a significant predictor of project-related complexities?" Keeping that in view, the current study depicted a significantly negative influence of implementing agile management on the project complexity. This suggests that the implementation of agile management enabled the regarding project managers to be able to effectively foresee the undertaken projects to a greater extent by adapting agile management practices than they would otherwise have had by adapting traditional management practices. The respective findings were found in alignment with the study conducted by Sohi, Hertogh [9] on 67 projects of complex nature, depicted that in a hybrid system with agile management practices coupled with traditional management approach was able to mitigate the magnitude of complexity faced by the regarding firms. In another subjective study conducted by Maylor and Turner [7] projected deduction being based upon 43 workshops and the opinion of 1100 managers. The results suggested an agile management approach as possibly the most effective approach to diminish the project complexity to commendable levels. Akin to which, in an extensive literature review conducted by Bergmann and Karwowski [70] also concluded the similar findings that adaptation of agile management is very effective in terms of mitigating the project related complexities and a accomplishing project outcomes.

The third research hypothesis stated, "Is the project complexity a significant predictor of project performance?" Keeping that in view, the current study depicted a significantly negative influence of project complexity on the overall performance of the projects. This suggests that the uncertainties faced by the project manager may hinder the accomplishment of the project. This would further possibly result in causing unnecessary delays, financial losses, overused employee efforts, working environment with moral, quality compromises, and unsatisfied clients. The respective findings were found in alignment with the study conducted by Floricel, Michela [71] on 81 projects deployed 5 across continents, depicted the possible negative impact of complexities on the overall performance of the organizations; that may be faced at each step of the development process. In another hybrid study conducted by Zhu and Mostafavi [8] on various senior project managers employed in the construction sector as well opinionated that complexities associated with organizations can deter the performance observed across their respective projects. Likewise, Luo, He [72] compile the opinion of 245 project managers that expressed the fact that project complexity can jeopardize the accomplishment of desired organizational outcomes. Therefore, their mitigation is a necessity for an organization to thrive.

**Table 13. Mediation analysis.**

|  | Original Sample (O) | Standard Deviation (STDEV) | T Statistics (\|O/STDEV\|) | P Values |
|---|---|---|---|---|
| **Agile Management > Project Complexity > Project Performance** | 0.104 | 0.049 | 2.144 | 0.032 |

**Table 14. Moderation analysis.**

|  | Original Sample (O) | Standard Deviation (STDEV) | T Statistics (\|O/STDEV\|) | P Values |
|---|---|---|---|---|
| **Moderating Effect > Project Performance** | 0.105 | 0.049 | 2.119 | 0.034 |

The fourth research hypothesis stated, "Are leadership competencies a significant predictor of project performance?" Keeping that in view, the current study depicted a significant relationship between leadership competencies and project performance. This suggests that effective leadership can play a pivotal role in enabling an organization to attain the desired performance targets associated to its respective project. The respective findings were found in alignment to the study conducted by Ahmed and Anantatmula [34] on 286 project managers serving various construction firms in Pakistan, suggested leadership competencies be an effective measure to enhance the performance of the projects it is utilized onto. In another hybrid study conducted by Berssaneti and Carvalho [73] on 336 project managers deployed across various Brazilian firms opinionated that effective supervision and managerial support can prove itself to be a potential factor in enabling a firm to deliver desired outcomes.

The fifth research hypothesis stated, "Does the project complexity mediate the relationship between agile management practices and project performance?" Keeping that in view, the current study depicted a significant relationship between agile management and project performance while considering leadership competencies as a moderator. This suggests that effective implementation of agile management practices in a project can prove themselves to be effective in elevating project performance. Though the magnitude of complexity associated with the project can explain the possible decline observed in project performance; regardless of the management practices being observed. Though the observed decline can be minimized to a laudable extent through the utilization of agile management practices. The respective findings were found in alignment with an in-depth correlational study conducted by Sohi, Hertogh [9] on 67 project managers supervising various projects. The results suggested that inducing agile management practices within any compatible system can enable an organization to manage through its professional challenges which can possibly lead an organization to perform better.

The sixth research hypothesis stated, "Do the leadership competencies moderate the relationship between agile management practices and project performance?" Keeping that in view, the current study depicted a significant relationship between project complexity and project performance while considering leadership competencies as a moderator. Which suggests that effective implication of human factor in terms of leadership competencies can play a vital role in mitigating the hindrances faced during the project development process and can further

**Table 15. Results summary.**

| Index | Hypothesis | Result |
|---|---|---|
| **H1** | Agile management practices will significantly impact the project performance, in a positive manner. | Accepted |
| **H2** | Agile management practices will significantly impact the project complexity, in a negative manner. | Accepted |
| **H3** | Project complexity will significantly impact project performance, in a negative manner. | Accepted |
| **H4** | Leadership competencies will significantly impact the project performance, in a positive manner. | Accepted |
| **H5** | Project complexity will significantly mediate the relationship between agile management practices and project performance. | Accepted |
| **H6** | Leadership competencies will significantly moderate the relationship between agile management practices and project performance. | Accepted |

result in enhanced performance. On the contrary, the absence of required leadership competencies can result in augmentation of adversities that may lead to a decline in the project performance. The respective findings were found in alignment to a mixed-mode study conducted by Aurélio de Oliveira, Veriano Oliveira Dalla Valentina [74] on 32 highly skilled and influential project managers in the field of R&D; who have served various forms globally. The correlational study depicted a possibly potential impact of an appropriate leadership approach to resolve organizational situations and deliver targeted performance.

## Conclusion

Considering the hypothetical contemplations of the current study, various deductions have been made. To begin with, the implementation of agile management practices in the Pakistani I.T industry proves itself to be effective in terms of enhancing the overall performance of the undertaken projects. Thus, ensuring the sustainability of organizations in the industry. Moreover, it was observed that agile management practices enabled its utilizers to cope up with the complexities, by breaking down tasks into smaller work units and implementing the supervision on a horizontal scale rather than top-down. This approach not only made managing tasks effectively and efficiently but also made the decision making swift. Though it was observed that the organizations that weren't able to take on the implementation of agile management practices on a full scale, faced complexities in various organizational terms, that would lead to declined performance. In addition to the mitigation of complexities through the implementation of agile management practices, it was the effective consideration of human factors in terms of leadership competencies that extended the reduction of organizational complexities and upscaled the magnitude of performance delivered.

The current study offers a pathway to understanding the application of agile management practices in the IT sector. Though it faces various shortcomings in both contextual and conceptual manner, which can further serve as a pathway to future researchers and professionals to look into and extend the knowledge pool.

In conceptual terms, the current study only took into account one mediatory variable i.e., project complexity to explain the implications of agile management onto the project performance. Akin to which, only one moderating variable was considered to evaluate the variability in the magnitude of project performance. Both of these are not enough of a consideration to depict the full potential of application of agile management practices in determining the project performance. Referred to which, it is commended for the future researchers and professionals to look into considering other variables that can explain the phenomena of agile management to variate the magnitude of project performance delivered. In alignment to which, it will also be interesting to see the implementation of agile management to enhance the organizational accomplishments such as, attaining competitive advantage, innovation, industrial sustainability, and more.

In contextual terms, the current study has targeted the IT sector of Pakistan; a developing nation. Since other industries as well are realizing the necessity of agile management and embracing its practices, it will be interesting to see the application of similar study in other developing nations, as well as other industrial sectors.

## Supporting information

**S1 Appendix.**
(DOCX)

**S1 Dataset.**
(CSV)

## Author Contributions

**Conceptualization:** Umer Muhammad, Ahsen Maqsoom.

**Data curation:** Umer Muhammad, Najam Muhammad, Ahsen Maqsoom, Khuram Shafi.

**Formal analysis:** Umer Muhammad, Khuram Shafi.

**Investigation:** Umer Muhammad, Khuram Shafi.

**Methodology:** Umer Muhammad, Najam Muhammad, Ahsen Maqsoom.

**Project administration:** Umer Muhammad, Tahira Nazir, Najam Muhammad, Ahsen Maqsoom, Samina Nawab, Syeda Tamkeen Fatima, Khuram Shafi.

**Resources:** Umer Muhammad, Samina Nawab, Syeda Tamkeen Fatima, Faisal Shafique Butt.

**Software:** Umer Muhammad, Faisal Shafique Butt.

**Supervision:** Tahira Nazir, Ahsen Maqsoom, Samina Nawab, Faisal Shafique Butt.

**Validation:** Umer Muhammad, Samina Nawab, Syeda Tamkeen Fatima, Khuram Shafi, Faisal Shafique Butt.

**Visualization:** Umer Muhammad.

**Writing – original draft:** Umer Muhammad, Najam Muhammad.

**Writing – review & editing:** Umer Muhammad, Tahira Nazir, Syeda Tamkeen Fatima, Faisal Shafique Butt.

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
