## [Decision Letter · Decision Letter 0]

11 Nov 2020

PONE-D-20-27419

IMPACT OF AGILE MANAGEMENT ON PROJECT PERFORMANCE: EVIDENCE FROM I.T SECTOR OF PAKISTAN

PLOS ONE

Dear Authors,

Thank you for submitting your manuscript to PLOS ONE. After careful consideration, we feel that it has merit but does not fully meet PLOS ONE’s publication criteria as it currently stands. Therefore, we invite you to submit a revised version of the manuscript that addresses the points raised during the review process.

 Please see comments below.

We look forward to receiving your revised manuscript.

Kind regards,

Dejan Dragan, PhD

Academic Editor

PLOS ONE

Additional Editor Comments (if provided):

The article was reviewed by two reviewers. They both require a major revision. Therefore, it is suggested that the authors strictly follow the instructions and comments of the reviewers.

Journal Requirements:

 [The funders had no role in study design, data collection and analysis, decision to publish, or preparation of the manuscript.].

5. We note you have included a table to which you do not refer in the text of your manuscript. Please ensure that you refer to Table 2, 3, 4, 5, 6, 7, 8, 9, 10, 11, 12, 13, 14, 15 and 16 in your text; if accepted, production will need this reference to link the reader to the Table.

6. Please include a copy of Tables 4.1, 4.2, 4.3, 4.4, 4.5, 4.6, 4.7, 4.8, 4.9, 4.10, 4.11, 4.12, 4.13 and 4.14 which you refer to in your text on pages 12, 13, 14, 15, 16, 17, 20, 21 and 22.

Reviewers' comments:

Reviewer's Responses to Questions

**Comments to the Author**

1. Is the manuscript technically sound, and do the data support the conclusions?

Reviewer #1: Yes

Reviewer #2: Yes

2. Has the statistical analysis been performed appropriately and rigorously? 

Reviewer #1: Yes

Reviewer #2: Yes

3. Have the authors made all data underlying the findings in their manuscript fully available?

Reviewer #1: Yes

Reviewer #2: Yes

4. Is the manuscript presented in an intelligible fashion and written in standard English?

Reviewer #1: Yes

Reviewer #2: Yes

5. Review Comments to the Author

Reviewer #1: The authors present an interesting and thorough analysis of how Agile Management plays a key role in the execution of IT projects. The authors have consulted numerous bibliographic sources. From the previous studies the authors have presented 6 hypotheses that they have tested by analyzing the information received in the surveys.

A representative sampling of the population to be studied has been made. A previous analysis of the reliability of the data has been made before starting the battery of tests that would allow testing the hypotheses. The authors have been very conscientious and have carried out numerous statistical tests trying to validate or reject the hypotheses presented.

According to the tests they have selected and developed, they accept the 6 hypotheses put forward, from which they develop the discussion. Some of the hypotheses may seem that they could be answered from a logical approach, however, it seems to me very appropriate and necessary that the authors have been able to give it a scientific support.

On the other hand, they recognize and detail the limitations of the study and even draw some conclusions. This seems to me very relevant, since in the conclusions they encourage the virtues of this approach and encourage third parties to apply them in the development of their projects. Therefore, it shows the confidence of the authors in their analysis and results. It is important, because in this case they have the capacity to experiment in the future. It would be very interesting if, in a few years, from the acceptance of the results of this study, the results of real experiences could be presented.

Taking into account that the concept has been used in the literature since 1991, I have missed in the discussion some concrete examples of projects that have been developed under this framework, their results, which could support the accepted hypotheses.

There are several aspects, especially structural, that do not convince me. Although the journal gives freedom to rename the sections, the structure of this publication is not clear to me.

Regarding question 3 of this questionnaire (Have the authors made all data underlying the findings in their manuscript fully available?), I suppose that, as usual in PlosOne, the results of the surveys will be available to the scientific community.

General

According to the rules of the magazine, up to three hierarchical levels can be reached in the headings. Authors sometimes go as high as a four. This is the case in STATISTICAL RESULTS & ANALYSIS (1), Structural Equation Modeling (2), SEM (Stage 1) (3) and Instrument's Reliability (4)

Although the authors' English is correct, I think they could use an external review to detect small errors and propose some grammatical changes.

LITERATURE REVIEW

After the introduction, there is a section that is Literature Review, with several sections, in which there are mixed elements that could fit both in the introduction, and in results and discussion. The introduction is, based on the existing literature, the space to assess what has been done so far and draw the main lines of the work presented. I think the authors could pass these descriptive parts to a longer introduction divided into several sections, ending with the presentation of the hypotheses.

Similarly, there are fragments that may be more appropriate in the discussion. Table 1 is not clear whether it is own production (better in results) or taken from other authors.

RESEARCH METHODOLOGY

The methodology is fragmented as in the following section (STATISTICAL RESULTS & ANALYSIS), the results of each test are presented with its methodology.

STATISTICAL RESULTS & ANALYSIS

The first paragraph is methodology.

Demographical Classification

The text of this section is the same as that presented in table 2. I recommend leaving the table and limiting the text to summarize the relevant aspects for the study.

It is necessary for the authors to clarify the headings to make it easier to follow the sections (this can be applied to the whole document).

There is no correspondence between the number of the tables in the text and the actual number of each table.

In general, I miss a descriptive text in the legend of the tables, beyond naming the test performed.

DISCUSSION

The authors have made a great statistical effort to support each of the hypotheses. I think it would be relevant to indicate which tests they have used to validate each of the hypotheses.

As I commented earlier, I believe that if the authors could identify real cases that would allow them to exemplify the hypotheses, they could strengthen their results. Similarly, I believe that, taking into account that there are 70 bibliographic references, it seems little to me that they only use 10 for discussion.

Reviewer #2: 1. The three main criteria for this manuscript are (a) quality and content of the research/review; (b) Quality, brevity and clarity of presentation; (c) Significance, relevance and timeliness of the topic. In addition, this title is (i) coverage of the literature/significant developments in the field or clarity of discussion within an emerging topic; (ii) originality, new perspectives or insights; (iii) international interest; and (iv) relevance for governance, policy or practical perspectives relevant to the focus of this manuscript. However, this study is lacking the most important criteria. Hence, I think the author needs to consider these criteria before your submission.

2 To be legible, the whole text must be completely edited with the help of a native English editor to polish your writing to prevent redundancies, grammatical errors and punctuation problems.

3. Please underscore the scientific value-added of your paper in your abstract and introduction.

4. Introduction should be clearly stated research questions and targets first. Then answer several questions: Why is the topic important (or why do you study on it)? What are the research questions? What has been studied? What are your contributions? Why is to propose this particular method? The outline of the paper can also be included. Please build upon the great work we have published on these subjects.

5. The major defect of this study is the debate or Argument is not clear stated in the introduction session. Hence, the contribution is weak in this manuscript. I would suggest the author to enhance your theoretical discussion and arrives your debate or argument.

6. The literature review is necessary for you to clarify the “contribution” of your study. In current form, there is none literature to support your study. The author failed to present the study debates and failed to discuss the debates. In general, the author should present a specific debate for your study.

7. In the Introduction and Literature review, the author conducts detailed literature discussions on agile management, but it would be better if the literature can be added in the last five years.

8. Please make sure your conclusions' section underscores the scientific value-added of your paper, and/or the applicability of your findings/results, as indicated previously. Please revise your conclusion part in more detail. Basically, you should enhance your contributions, limitations, underscore the scientific value-added of your paper, and/or the applicability of your findings/results and future study in this session.

6. PLOS authors have the option to publish the peer review history of their article (what does this mean?). If published, this will include your full peer review and any attached files.

Reviewer #1: **Yes: **Dr. Juan Jesús Bellido López

Reviewer #2: No

---

## [Author Response · Author response to Decision Letter 0]

26 Feb 2021

NOTE FROM AUTHORS: We are grateful to the editor and the reviewers for providing our research team with enlightening comments to improve our manuscript upon. We have incorporated all the suggested changes in alignment to the scope of the presented study.

EDITOR’S COMMENTS

COMMENT 1: Please ensure that your manuscript meets PLOS ONE's style requirements, including those for file naming. The PLOS ONE style templates can be found at

RESPONSE: Considering the manuscript preparation instructions, the manuscript has been reformatted. 

COMMENT 2: Please include additional information regarding the survey or questionnaire used in the study and ensure that you have provided sufficient details that others could replicate the analyses. For instance, if you developed a questionnaire as part of this study and it is not under a copyright more restrictive than CC-BY, please include a copy, in both the original language and English, as Supporting Information.

RESPONSE: Along with the complete information of the survey conducted for the current study in the methodology section, the revised draft now also contains the copy of the adapted survey questionnaire.

COMMENT 3: Thank you for stating the following financial disclosure. If you did not receive any funding for this study, please state: “The authors received no specific funding for this work.”

RESPONSE: The statement regarding the financial disclosure has been added in the revised draft. 

COMMENT 4: Please amend your list of authors on the manuscript to ensure that each author is linked to an affiliation. Authors’ affiliations should reflect the institution where the work was done (if authors moved subsequently, you can also list the new affiliation stating “current affiliation….” as necessary).

RESPONSE: The author list has been corrected.

COMMENT 5: We note you have included a table to which you do not refer in the text of your manuscript. Please ensure that you refer to Table 2, 3, 4, 5, 6, 7, 8, 9, 10, 11, 12, 13, 14, 15 and 16 in your text; if accepted, production will need this reference to link the reader to the Table.

RESPONSE: The table numbers and their in-text references have been updated in alignment to one another. Therefore, eliminating the ambiguity found in the previously submitted draft.

COMMENT 6: Please include a copy of Tables 4.1, 4.2, 4.3, 4.4, 4.5, 4.6, 4.7, 4.8, 4.9, 4.10, 4.11, 4.12, 4.13 and 4.14 which you refer to in your text on pages 12, 13, 14, 15, 16, 17, 20, 21 and 22.

RESPONSE: The table numbers and their in-text references have been updated in alignment to one another. Therefore, eliminating the ambiguity found in the previously submitted draft.

REVIEWER 1 COMMENTS

COMMENT 1: Taking into account that the concept has been used in the literature since 1991, I have missed in the discussion some concrete examples of projects that have been developed under this framework, their results, which could support the accepted hypotheses.

RESPONSE: The early implementation of agile management was mostly seen in the manufacturing sector as mentioned in the presented manuscript. Since, the current study focuses on the I.T sector, the authors have referred to the most recent studies regarding implementation of agile management in the I.T sector. These studies can be referred to in the introduction, literature review as well as the discussion section. The inclusion of the latest research regarding the variables included in the current study, enabled the authors to deliver the most relevant and most up-to-date information about the research area covered under scope of the current study.

COMMENT 2: There are several aspects, especially structural, that do not convince me. Although the journal gives freedom to rename the sections, the structure of this publication is not clear to me.

RESPONSE: The submitted manuscript mainly comprises of Introduction, Literature Review, Methodology, Results & Analysis, Discussion, & Conclusion sections now. Considering the PLOS One formatting guidelines the manuscript has been re-formatted to bring more clarity to the presented content and enhance its readability.

COMMENT 3: Regarding question 3 of this questionnaire (Have the authors made all data underlying the findings in their manuscript fully available?), I suppose that, as usual in PlosOne, the results of the surveys will be available to the scientific community.

RESPONSE: Yes, all the data has been made available.

COMMENT 4: According to the rules of the magazine, up to three hierarchical levels can be reached in the headings. Authors sometimes go as high as a four. This is the case in STATISTICAL RESULTS & ANALYSIS (1), Structural Equation Modeling (2), SEM (Stage 1) (3) and Instrument's Reliability (4).

RESPONSE: Considering the comment, the textual hierarchical levels have been reduced to three for enhanced readability experience.

COMMENT 5: Although the authors' English is correct, I think they could use an external review to detect small errors and propose some grammatical changes.

RESPONSE: The manuscript has been proofread and updated accordingly. 

COMMENT 6: After the introduction, there is a section that is Literature Review, with several sections, in which there are mixed elements that could fit both in the introduction, and in results and discussion. The introduction is, based on the existing literature, the space to assess what has been done so far and draw the main lines of the work presented. I think the authors could pass these descriptive parts to a longer introduction divided into several sections, ending with the presentation of the hypotheses.

Similarly, there are fragments that may be more appropriate in the discussion. Table 1 is not clear whether it is own production (better in results) or taken from other authors.

RESPONSE: The introduction section of the present research comprises of the background related to the opted variables. Moreover, the first half of the introduction section addresses the three conceptual research gaps that the present study aims to address. The following discussion addressed the contextual research gap the current study has addressed. While in the concluding paragraphs the research questions the current study will be able to resolve are stated. Lastly, the introduction section has been updated with the brief overview of the following section in the presented research.

In particular to Table 1, the stated differences have been based upon the prior research that now have been quoted alongside the heading of the table.

COMMENT 7: The methodology is fragmented as in the following section (STATISTICAL RESULTS & ANALYSIS), the results of each test are presented with its methodology.

RESPONSE: The methodology section discusses the adapted approach and measures to collect the data, as well as the overview of the statistical approach opted for the current study. In the following section, Results & Analysis the authors have discussed all the parameters that were opted to test the quality of the data and opted measures along with the significance of the proposed hypothesis. The individual explanation of each parameter was placed in the Results & Analysis section for the better understanding of the readers regarding the reason to opt each of the evaluation criteria as well as its respective threshold level. Merging the regarding explanations into the Methodology section would have resulted confusion for the prospect readers in understanding the results of the current study.

COMMENT 8: The first paragraph is methodology.

Demographical Classification

The text of this section is the same as that presented in table 2. I recommend leaving the table and limiting the text to summarize the relevant aspects for the study.

It is necessary for the authors to clarify the headings to make it easier to follow the sections (this can be applied to the whole document).

There is no correspondence between the number of the tables in the text and the actual number of each table.

In general, I miss a descriptive text in the legend of the tables, beyond naming the test performed.

RESPONSE: Considering the comment Table 2 has been removed.

The whole document has been reformatted to redefine and highlight the main sections of the study.

The table numbers and their in-text references have been revised.

The descriptive texts for corresponding legends used in table 5,7,8 and 9 have been added.

COMMENT 9: The authors have made a great statistical effort to support each of the hypotheses. I think it would be relevant to indicate which tests they have used to validate each of the hypotheses.

RESPONSE: The current study used path analysis in correspondence to their p-values to validate the hypothesis related to direct effect. While mediation and moderation effects along with their p-values were utilized to validate the mediation and moderation phenomena. The explanation associated to the validation of results have been provided in the path analysis, mediation and moderation test sub-sections. Finally, all the tested hypotheses have been stated in the Results Summary table.

COMMENT 10: As I commented earlier, I believe that if the authors could identify real cases that would allow them to exemplify the hypotheses, they could strengthen their results. Similarly, I believe that, taking into account that there are 70 bibliographic references, it seems little to me that they only use 10 for discussion.

RESPONSE: The development of introduction and literature review has been based upon the real-case studies conducted in various contexts. Furthermore, the discussion related to each of the proposed hypothesis and their significance in the discussion section has been explained in alignment to the more of studies based upon the real-case scenarios.

In particular to the discussion section, more relevant studies have been added to strengthen the presented facts.

The authors have tried to built the introduction, literature review and discussion section upon most recent studies to assure that the presented manuscript represent the most updated knowledge on the subject.

REVIEWER 2 COMMENTS

COMMENT 1: The three main criteria for this manuscript are (a) quality and content of the research/review; (b) Quality, brevity and clarity of presentation; (c) Significance, relevance and timeliness of the topic. In addition, this title is (i) coverage of the literature/significant developments in the field or clarity of discussion within an emerging topic; (ii) originality, new perspectives or insights; (iii) international interest; and (iv) relevance for governance, policy or practical perspectives relevant to the focus of this manuscript. However, this study is lacking the most important criteria. Hence, I think the author needs to consider these criteria before your submission.

RESPONSE: Thank you for the kind comment. Considering the instructions, the authors have revised the manuscript in terms of establishing the significance of the presented research across all the relevant sections of the current research. Moreover, all the sections of the presented manuscript have been aligned with one another for clarity and enhanced readability.

COMMENT 2: To be legible, the whole text must be completely edited with the help of a native English editor to polish your writing to prevent redundancies, grammatical errors and punctuation problems.

RESPONSE: The complete manuscript has been proofread and revised accordingly.

COMMENT 3: Please underscore the scientific value-added of your paper in your abstract and introduction.

RESPONSE: The conceptual and contextual contribution of the current study has been added in the abstract as well as the introduction section.

COMMENT 4: Introduction should be clearly stated research questions and targets first. Then answer several questions: Why is the topic important (or why do you study on it)? What are the research questions? What has been studied? What are your contributions? Why is to propose this particular method? The outline of the paper can also be included. Please build upon the great work we have published on these subjects.

RESPONSE: The introduction section is built upon the prior studies that justify the importance of the agile management in attaining optimum project performance and its importance in the current study. Further each paragraph of the introduction section addresses the conceptual research gap associated to each opted variable. Moreover, the contextual research gap of the current study has been addressed as well. In the closing paragraphs the addressed research question of the current research have been added. Lastly, the overview of the following sections in the paper has been updated as well.

COMMENT 5: The major defect of this study is the debate or Argument is not clear stated in the introduction session. Hence, the contribution is weak in this manuscript. I would suggest the author to enhance your theoretical discussion and arrives your debate or argument.

RESPONSE: The introduction section of the present research comprises of the background related to the opted variables. Moreover, the first half of the introduction section addresses the three conceptual research gaps that the present study aims to address. The following discussion addressed the contextual research gap the current study has addressed. While in the concluding paragraphs the research questions the current study will be able to resolve are stated. Lastly, the introduction section has been updated with the brief overview of the following section in the presented research.

COMMENT 6: The literature review is necessary for you to clarify the “contribution” of your study. In current form, there is none literature to support your study. The author failed to present the study debates and failed to discuss the debates. In general, the author should present a specific debate for your study.

RESPONSE: The presented manuscript includes a detailed literature review regarding each of the variable included in the study. Moreover, the supporting studies supporting the hypothetical development of each argument has also been included. Further in the revised document the literature review has been updated with the latest references to strengthen the basis of the conducted research.

COMMENT 7: In the Introduction and Literature review, the author conducts detailed literature discussions on agile management, but it would be better if the literature can be added in the last five years.

RESPONSE: Considering the comment, the literature review has been updated with the latest references.

COMMENT 8: Please make sure your conclusions' section underscores the scientific value-added of your paper, and/or the applicability of your findings/results, as indicated previously. Please revise your conclusion part in more detail. Basically, you should enhance your contributions, limitations, underscore the scientific value-added of your paper, and/or the applicability of your findings/results and future study in this session.

RESPONSE: The conclusion section includes the explanation related to the conceptual and contextual gaps addressed by the presented research. Moreover, the potential shortcoming associated to the present research as well as the future suggestion to address those limitations have also been updated.

---

## [Editor Report · Decision Letter 1]

16 Mar 2021

IMPACT OF AGILE MANAGEMENT ON PROJECT PERFORMANCE: EVIDENCE FROM I.T SECTOR OF PAKISTAN

PONE-D-20-27419R1

Dear Authors,

We’re pleased to inform you that your manuscript has been judged scientifically suitable for publication and will be formally accepted for publication once it meets all outstanding technical requirements.

Kind regards,

Dejan Dragan, PhD

Academic Editor

PLOS ONE

Additional Editor Comments (optional):

All the reviewers' comments have been adequately addressed. Accordingly, the acceptance of the paper is recommended.
---

## [Editor Report · Acceptance letter]

25 Mar 2021

PONE-D-20-27419R1 

Impact of Agile Management on Project Performance: Evidence from I.T sector of Pakistan 

Dear Dr. Muhammad:

I'm pleased to inform you that your manuscript has been deemed suitable for publication in PLOS ONE. Congratulations! Your manuscript is now with our production department. 

Kind regards, 

on behalf of

Dr. Dejan Dragan 

Academic Editor

PLOS ONE